# A Systematic Review Exploring the Theories Underlying the Improvement of Balance and Reduction in Falls Following Dual-Task Training among Older Adults

**DOI:** 10.3390/ijerph192416890

**Published:** 2022-12-15

**Authors:** Mohammad Jobair Khan, Priya Kannan, Thomson Wai-Lung Wong, Kenneth N. K. Fong, Stanley John Winser

**Affiliations:** Department of Rehabilitation Sciences, The Hong Kong Polytechnic University, Kowloon, Hong Kong 999077, China

**Keywords:** motor cognitive interference, postural control, falls, older adults

## Abstract

Background: Balance impairment causes frequent falls in older adults, and preventing falls remains challenging. Dual-task (DT) training reduces falls by improving balance, but the precise theory is not fully understood. This review aims to explore the theories underlying the effectiveness of DT in improving balance and reducing falls in older adults. Methods: Eleven electronic databases were searched from database inception to June 2022. Two reviewers independently performed study screening and data extraction. The risk of bias (RoB) in the included studies was assessed using the Cochrane Collaboration RoB 2 tool. Results: The searches yielded 1478 citations, of which 30 studies met the inclusion criteria and were included in the review. Twenty-two of the 30 included studies utilized the motor-cognitive type of DT for training, while six used motor-motor and two utilized cognitive–cognitive DT. The included studies reported 20 different theories to explain the effectiveness of DT for improving balance and reducing falls in older adults. The predominant theory identified in the included studies was attention theory (*n* = 14). Overall, 26 studies reported improved balance and five studies found a reduction in fall incidence following DT training. Balance and falls improved significantly in 15 motor-cognitive DT intervention studies. Conclusion: Attention shifting between two tasks is reported to occur following DT training. Motor-cognitive DT training improves balance and reduces fall incidence in older adults by shifting attention based on the difficulty and priority of a task from the motor to the cognitive task.

## 1. Introduction

Globally, falls is the leading cause of unintentional death among older adults [1]. Moreover, falls is the 18th-leading cause of disability-adjusted life years in older adults [2,3]. Though the global incidence and prevalence of falls have dropped (by 3.7% and 6.5%, respectively) over the past two decades [4], managing falls remains a challenge among older adults. A common risk factor for falls is balance impairment [5]. One in five older Americans had balance impairment, and the condition was more common in women than in men [6]. One in four people with impaired balance has difficulty with daily activities [6]. Approximately 10% of falls result in fractures, which are also a significant source of morbidity and mortality in older adults [7,8]. Falls decrease quality of life and confidence, increase fear of falling, and limit functional ability and interpersonal interactions [9,10]. Even minor fall-related injuries are reported to cause pain, limited function, and high medical bills [7].

The causes of falls are multifactorial [11]. Poor balance is one of the crucial contributors and can result from white matter lesions in the brain that frequently lead to a decrease in multitasking activity [12,13,14,15]. White matter lesions have been linked to motor and cognitive dysfunction during multitasking, which degrades balance performance [15,16]. In multitasking, interference between two tasks reduces performance in one or both tasks [17], and this interference occurs between motor and cognitive tasks [18]. Thus, many studies [19,20] used dual-task (DT) paradigms to investigate how balance and cognition interact.

Over the past two decades, there has been a growing literature [21,22,23] on DT training, which entails the concurrent performance of a primary and a secondary task [19]. Initial attempts at performing DT are challenging, as it involves responding to two stimuli [24]: The primary task corresponds to the first delivered stimulus, whereas the secondary task corresponds to the second stimulus [24]. Secondary task response time is typically slower than that of the primary task [24]. This could be due to the longer processing time for initiating the secondary task [24]. With practice, the processing time of the secondary task shortens, and it becomes easier to perform DT [19]. The DT paradigm is relevant because most daily tasks require simultaneous cognitive and motor performance [19].

Researchers have proposed plausible, convincing, and credible theories to explain the theories by which DT training in a study might improve posture and attention, turning, gait, and gait inhibition [25,26,27]. According to attention theory, DT promotes balance by improving attention-shifting between two tasks [28]. In contrast, another theory hypothesized that the multicomponent training approach is useful for developing balance because it includes different exercises targeting different cognitive functions [29]. Proposed theories are effective because they enhance balance control in older adults [28,29,30].

This systematic review provides an overview of the theories that have been proposed to underlie the effectiveness of DT in improving balance and reducing falls in older adults in randomized control trials (RCTs). A previous review evaluated the theories underpinning motor-cognitive interference and balance and gait among healthy young adults [25]. A recent systematic review [31] identified DT training as an effective strategy for improving balance and reducing falls in older adults. However, to the best of our knowledge, there are no reviews exploring the theories underlying the effectiveness of DT training in improving balance and reducing falls among older adults. A better understanding of the underlying theories would benefit researchers by helping them plan appropriate interventional studies and informing clinical decision-making based on a summary of dosages reported in studies. Familiarity with the theories proposed in studies [32] to explain the association between balance impairment and older adults is important in the design of effective interventions using DT to prevent falls in this population. Thus, this review aims to explore the theories by which DT training improves balance and reduces falls among older adults.

## 2. Materials and Methods

This systematic review was developed and is reported in line with the PRISMA guidelines (see Appendix B, Table A4) [33]. The review was prospectively registered with PROSPERO (Ref No.: CRD42022315998; https://www.crd.york.ac.uk/prospero/#searchadvanced, accessed on 30 May 2022).

### 2.1. Search Process

Multiple electronic databases (PubMed, MEDLINE, Cochrane, CINAHL, Embase, Web of Science, Scopus, PsycInfo, PEDro, CNKI, and Wanfang) were searched from database inception until 27 June 2022. Hand searches were also conducted among the reference lists of the included studies. We constructed five search themes: DT, balance, fall, older adults, and RCT. Search terms were specific to each database. Appendix A Table A1 reports the search strategy for the PubMed database. The Boolean expression “OR” was used to combine relevant terms under each theme and “AND” was used to combine the five themes. Keywords in the searches used that did not match MeSH phrases. The root words “Posture” and “RCT” were truncated.

### 2.2. Eligibility Criteria

To be included, the studies had to (1) have included older adults (aged >60 years) without any pathological conditions; (2) have delivered DT as the intervention of interest; (3) have explained the underlying theory for the improvement of balance or reduction in falls following DT training; (4) be RCTs or pilot RCTs with a randomized cluster or cross-over design; (5) have utilized any of the following outcome measures for evaluating balance: Berg Balance Scale (BBS), Timed Up and Go (TUG), Community Balance and Mobility Scale (CB&M), one-leg stance (OLS), tandem test (TT), Romberg test (RT), Step Test (ST), Fullerton Advanced Balance (FAB), four square step test (FSST), Figure of 8 Walk Test (F8W), Frailty and Injuries Cooperative Studies of Intervention Techniques test (FICSIT-4), Y-balance test (Y-BT), Tinetti Balance Assessment Tool Balance Exercise (TBAT), Tinetti performance-oriented mobility score (TPOMA), and Functional Reach Test (FRT); and (6) utilize the number, percentage, and incidence rate ratio for evaluating falls. Unpublished theses were also included in the review. Studies were ineligible if they examined the combined effects of DT training with therapies such as dance, drugs, music, karate, tai chi, and brain stimulation because these interventions might interfere with the effect of DT training [34,35]. Studies with unavailable full text, study protocols, conference abstracts, and studies without reliable and valid scales used for balance measurement were excluded [36,37,38]. The validity of a scale was primarily defined by the sample of participants to ensure that the outcomes were applicable to a diverse array of demographics, cultures, and other contexts [39]. Studies published in languages other than English and Chinese were also excluded. The reliability of the scale was determined by the consistency of the outcome [39].

### 2.3. Data Extraction

Two independent reviewers screened titles, abstracts, and full texts. A third reviewer was consulted to resolve discrepancies between the reviewers. Data were extracted using a standard form. Extracted data included study characteristics, intervention types, dosage, treatment effects, and proposed theories.

### 2.4. Risk of Bias

The Cochrane Collaboration Risk of Bias (RoB 2) tool was used to assess bias [40]. The RoB 2 tool analyzes randomization, intervention deviations, missing outcome data, outcome measurement bias, and result reporting bias. Each question was answered with “yes”, “probably yes”, “probably no”, “no”, or “no information” [40]. The bias risk of each domain was rated as “low”, “some concerns”, or “high.” Similar to the individual domains, overall RoB 2 was also summarized as “low”, “some concerns”, or “high” risk of bias [40].

### 2.5. Statistical Analysis

We calculated the agreement between the two authors using the kappa value for the data screening process and quality appraisal. Values ≤ 0 indicated no agreement; 0.01–0.20 indicated no to little agreement; 0.21–0.40 indicated fair agreement; 0.41–0.60 indicated moderate agreement; 0.61–0.80 indicated substantial agreement; and 0.81–1.00 indicated nearly perfect agreement [41]. The RoB 2 tool was used to assess the bias and the methodological quality of specific results of RCTs. Since the focus of this review was to explore the theories, not the treatment effects, a quantitative analysis such as a meta-analysis or meta-regression was not considered necessary.

## 3. Results

### 3.1. Search Results

The electronic searches yielded 1478 potentially relevant studies. Figure 1 summarizes the flow of studies through the review. After a stepwise screening process, 32 studies were found to be eligible for review. Two studies [42,43] were excluded after screening for full text. Two studies were excluded as they did not report any theories for explaining the treatment benefits of DT training. Therefore, this review included 30 studies. The agreement between the two review authors was near perfect (0.92) for full-text screening. Studies excluded at the full-text screening stage and the reasons for exclusion are reported in Appendix A Table A2.

### 3.2. Risk of Bias

The findings of the RoB assessment are illustrated in Figure 2. Overall, there was a low to moderate RoB across more than 38.5% of the studies. Twenty-six percent (*n* = 8) of the included studies were at low risk of bias, while 12.5% (*n* = 4) of the studies drew “some concerns” about outcome measurement, randomization, and deviation from intended intervention. “High” RoB was identified in 60% of studies (*n* = 18). The major methodological flaws were identified in measuring outcomes [23,44,45,46,47], missing outcome data [29,48,49], or both [22,50,51,52,53]. For methodological flaws for measuring outcome, firstly, flaws resulted because there was insufficient information available about whether outcome assessors were aware of the intervention that study participants had received. Secondly, the influence of the knowledge of the intervention on the assessment was addressed inadequately. For missing outcome data, the studies were reported “no, possibly no, or no information” if either the missing data was not reported or the statistical analysis for handling the missing data was not clearly reported.

### 3.3. Characteristics of Included Studies

#### 3.3.1. Types of Dual-Task Training

The DT interventions delivered in the included studies were (1) motor-cognitive, (2) motor-motor, or (3) cognitive–cognitive DT, as reported in Table 1. Among the 30 included studies, 22 studies [21,22,29,44,46,47,48,49,50,51,52,53,54,55,56,57,58,59,60,61,62,63] used motor-cognitive DT training, 6 [23,28,30,45,59,64] used a motor-motor type of task, and 2 [65,66] included cognitive exercise in both tasks.

#### 3.3.2. Types of Exercises

The demographics, types of DT interventions, treatment dosage, effectiveness, outcome measurement, authors’ conclusion, and theories of action proposed by the authors from the 30 included studies are reported in Table 2. Further detailed descriptions of the exercises, dosage of primary and secondary tasks, and control groups from the included studies are reported in Appendix A Table A3. Across the 30 included studies, different types of exercises were applied. Balance exercises appeared most frequently, as they were used in 12 studies [21,22,30,47,50,51,54,58,61,62,63,64]. Balance exercises were performed using free-hand, low-tech systems with minimal technology support or technology-dependent, computerized balance systems. Balance exercise combined with resistance exercise was performed in four additional studies [48,49,57,60].

#### 3.3.3. Dosage of Dual-Task Training

Training sessions lasted between 3 [45] and 90 min [30] per session. Eleven studies [22,23,30,44,49,51,55,59,61,64,65,66,67] had 60 min or longer sessions. The frequency of training varied from once a week to every day and the length of training ranged from four weeks to a year [44,45,59,60]. Notably, in one study [45], twice-daily exercise lasted only three minutes. Only seven studies [28,44,45,57,63,64,65] reported the duration of the secondary task, and the highest duration noted was 30 min [63].

#### 3.3.4. Study Comparator and Follow-Up

Thirteen of the 30 included studies [21,23,30,45,47,50,51,52,54,64,65,66,67] employed passive comparators instructing participants to either follow the fall prevention booklet (*n* = 1), continue with routine care (*n* = 3), or receive no intervention (*n* = 9). One study did not specifically report pre- and post-assessment time [44]. Six studies [23,43,48,51,56,57,58,59,60,61,62,63,64,65,66,67] included a follow-up assessment after the intervention phase, and the follow-up period ranged from 2 weeks [48,51] to 12 months [57].

#### 3.3.5. Outcome Measures

Balance was assessed in 25 studies [21,22,23,28,29,44,45,46,47,48,49,50,52,53,54,55,58,59,60,61,62,63,64,65,66]. Almost 30% of the studies employed the BBS [23,28,50,51,55,58,59,63,66], while six [45,49,52,60,62,65] used the TUG scale, and both scales were utilized in three studies [46,47,61]. Three studies [30,56,57] examined falls and reported the data as a percentage [57] or incidence rate ratio [30,56]. Both outcomes, balance and falls, were measured in two studies [51,67].

#### 3.3.6. Treatment Effects

Twenty-three (76.6%) [21,22,23,28,29,44,45,46,47,48,50,51,52,53,55,58,59,61,62,63,64,65,66] of the 30 included studies reported improvement in balance after DT training, while 5 (17.8%) [30,51,56,57,67] reported a reduction in the number of falls. Seventeen (63%) [21,22,28,29,44,46,48,51,52,55,59,61,62,63,64,65,66] out of 27 studies found significant improvement in balance measured using the BBS, TUG, TBAT, FAB, FSST, F8W, FRT, CM&M, FR, or Y-BT scales and 1 (20%) [56] out of 5 studies reported significant improvement in falls using the incidence rate ratio. Fifteen motor-cognitive training studies [21,22,23,28,29,44,46,48,51,52,55,61,62,63,64] demonstrated significant balance improvement after DT training, whereas three [28,59,64] out of six [23,28,45,47,59,64] studies using motor-motor and two studies using cognitive–cognitive [65,66] types of training demonstrated significant balance improvement as assessed using the BBS, TBAT, F8W, CB&M, FR, T-BT, TUG, FSST, and FRT scales.

#### 3.3.7. Theories Reported in the Included Studies

Among the 30 included studies, 20 different theories to explain balance improvement following DT interventions were identified. A single theory was documented in 19 studies [22,28,29,30,44,45,46,47,48,49,51,53,56,59,60,63,64,65,67]. More than one theory was reported in 12 studies [21,23,50,52,54,55,56,57,58,61,62,66], and of these, four studies [23,52,58,61] reported four theories. The attention theory was proposed to explain the improvement in balance and reduction in falls following the DT intervention in 14 studies [22,23,28,44,45,46,48,51,58,59,60,62,64,66]. The predominant theory invoked was the competition theory of attention, which was used in over a third [22,28,44,45,46,48,51,58,59,60,64] of all studies primarily (explained first, either alone or along with other theories) and in three studies secondarily (not explained as a primary theory, but presented along with other theories) [23,62,66]. The competition theory of attention has been extensively proposed as an underlying theory to explain improving balance, mostly in the application of motor-cognitive and motor-motor types of DT training. Figure 3a illustrates the pathway for improvement in balance following DT intervention using the attention theory model. In this model, the attention of the brain responds to a winning stimulus from multiple competing stimuli. In the context of DT, for example, the winning stimulus of the calculation or cognitive task draws attention during the balancing exercise or motor task. This shifting attention works to improve motor-cognitive capabilities, which contributes significantly to the improvement in balance. DT theories commonly suggested to improve balance were executive function in six studies [23,54,55,57,65,66] and a multicomponent training approach in five studies [29,30,50,55,67]. The executive function theory focuses on enhancing cognition to improve balance by loading on inhibiting, updating working memory, and task-set shifting [68], as illustrated in Figure 3b, while multicomponent theories targeting procedural memory work via multi-phase cognitive functions to improve balance [69,70], as shown in Figure 3c. The working memory model was invoked as a primary [21,49] or secondary theory [23,52] in two studies each. The working memory model focuses on neural efficacy to improve balance via the management of multiple task processes [23,71], as described in Figure 3d. Likewise, the divided attention theory, which focuses on rapidly shifting or splitting attentional focus between two tasks, was referred to as a primary [23,52] or secondary theory [23,52,57,58] in two and four studies, respectively.

## 4. Discussion

To the best of our knowledge, this is the first systematic review to summarize the possible theories underlying the effectiveness of DT interventions for improving balance and reducing falls in older adults. Identifying the spectrum of theories proposed by authors underlying the improvement of balance following DT was the objective of this review. Interestingly we found there were at least 20 theories explaining the improvement following the intervention. We notice that most of the studies reported more than one theory for the identified benefits. This is an indication that DT results in the improvement of multiple domains such as attention, procedural memory executive function and motor function.

The 30 included studies, filtered from a total of 1478 studies identified via database searches, proposed 20 distinct DT theories. These proposed theories were reported to explain the improvement in balance (*n* = 26) and the reduction in the number of falls (*n* = 5) following DT intervention among healthy older adults. The most frequently proposed theory was the competition theory of attention, which describes the shifting of attention from one task to a secondary task with DT practice. The DT interventions improved both motor and cognitive function compared to various controls.

The competition theory of attention was used to describe the improvement in balance and reduction in falls following DT training; it was cited primarily in 11 studies (36.7%) and secondarily in three additional studies (10%). This theory is derived from the definition of attention first proposed by William James in 1890 [72] and involves the central nervous system, where neurons are subjected to a wide range of internal and external stimuli at any given instance. Each stimulus competes for the attention of the nervous system [73]. The nervous system filters the stimuli by considering the challenges and attending to the prioritized task [74]. Based on the difficulty and priority of a task, attention shifts from one task to another. For instance, when an individual is sitting on an inflatable exercise ball and catching a tennis ball simultaneously, attention needs to shift from catching the tennis ball to sitting on the inflatable exercise ball to ensure the safety of the user. The attention theory was used to explain balance improvement, as the participants were able to attend to two simultaneous tasks efficiently with repeated practice, and the number of falls was reduced due to their capacity to increase their multi-tasking ability. Talwer et al. [59] delivered a square stepping exercise that involved switching one’s attention while passing a ball under variable priority instructions. The significant difference in balance measurement after training revealed that gradual practice decreases the reaction time of passing balls [59].

The competition theory of attention led to the development of the divided attention theory and the selective attention theory. These theories are mentioned in four studies [23,52,57,58]. The divided attention concept addresses the limitations of multitasking in information processing. The inability to process all information simultaneously demands division of attention, which splits or rapidly switches the attentional focus [75]. DT results in enhanced divided attention with the use of optimal attention resources in motor control [19]. This enables splitting attention between the primary and secondary tasks, which improves the outcomes of the functional task. The selective attention theory explains how multiple dynamic events and static sources of input are filtered and subsequently perceived, cognitively processed, and ultimately responded to [76,77]. Balance is considered a dynamic event [78], and the motor-cognitive DT interventions were intended to improve multiple dynamic events, including balance [78]. This intervention promotes parallel information processing and manipulating information [76]. Selective attention to task-relevant inputs and decision-making about balance performance leads to balance improvement [78,79]. One study [52] reported that motor-cognitive DT intervention was effective in improving balance and invoked the theories of both the divided and the selective attention theories as an explanation. Three more studies invoked the divided attention theory as an explanation for the improvement in balance [23,58] and reduction in falls [57] following DT intervention.

The multicomponent or multimodal training approach was proposed to explain the reduction in fall incidence following DT intervention in five studies [29,30,50,55,67]. This theory emphasizes that the DT intervention could activate procedural memory, which in turn enhances cognitive function [70]. Procedural memory is crucial for activities of daily life [80] and plays a role in multi-phase motor and cognitive functions. Simultaneous activity of the striatum and caudate nucleus improves procedural memory, targeting both motor and cognitive functions in one training program [69]. Thus, a multicomponent training program is appropriate for procedural memory improvement [70]. Based on this theory, the DT interventions are hypothesized to improve cognitive function, attention, cognitive control, memory, reasoning, and executive functions simultaneously [67,70]. Multiphase-based DT intervention improves specific functions by targeting balance through repeated exercises [81]. Repeated processes enhance cognitive function, improve balance [29], and reduce the number of falls [30] in older adults. One study [67] examined both outcomes and found reduced falls, but not improved balance for tandem scale. This may be because this study used the infrequently tandem balance measurement scales, which are not appropriate tests for aging participants (mean age of 81.9 years).

The executive function theory was reported in six studies [23,54,55,57,65,66]. It covers a spectrum of three key cognitive processes for improving balance by enhancing cognition: increasing the load on task inhibition, updating working memory, and task-set shifting [68,82]. Improvement is achieved through motor-cognitive DT training, DT exercise increases the load on inhibiting primary tasks or motor tasks, updates it on the working memory, and eases the motor-to-cognitive task switching [54,68]. Together, they improved executive function, which improved balance and reduced falls. Four studies [23,55,65,66] used this theory to explain an improvement in balance, while one study used it to explain a reduction in falls [57] following DT intervention in older adults.

Two additional theories based on the executive function theory have been proposed to explain how the motor-cognitive type of DT training helps to improve balance: the working memory model and the cognitive flexibility theory.

Cognitive flexibility theory explains the ability to spontaneously rearrange information in an adaptive response to substantially altering situational demands [83]. Task-switching activities, including DT, require cognitive flexibility [84]. This flexibility is achieved in DT exercises, which activate the prefrontal, anterior cingulate, and posterior parietal cortices and basal ganglia, enhancing cognitive flexibility [84]. This theory was proposed in one study [52] that demonstrated substantial improvement in balance after DT training.

Working memory is necessary for complex task processes, including reasoning, comprehension, and learning [71]. The working memory model was proposed to explain the results in four different studies [21,23,49,52]. Balance improvement occurred due to the neuronal efficacy resulting from the motor-cognitive DT training, which is responsible for comprehensive learning via amplifying cognition [23].

Six theories in three studies explained how DT performance improves balance and reduces falls. In one study [58], the following three theories were described: (1) a task-oriented approach emphasized improving balance or movement strategies within a given environment using motor-cognitive exercise [85,86]; (2) a task-automation model led to complete task automation [87]; and (3) a task integration model introduced effective integration of two tasks, minimizing the sharing capacity to improve DT performance [87]. These three models propose that balance is enhanced by automating one of the two DT exercises. Silsupadol et al. combined one of the previously automated tasks into fixed or variable priority instruction, with balance as the primary task and cognitive task training as the secondary task [58]. This increased the balance function in older adults with repeated practice.

In addition to the theories described above [58], three studies proposed the integrated motor and cognitive theory, the theory of reduced resources, the capacity-sharing theory, and the limited resource theory to explain the balance improvement following DT intervention. One study [56] evaluating the advantages of motor-cognitive DT training invoked an integrated motor and cognitive theory to explain the theory of fall reduction. According to this theory, motor, cognitive, and physical development are all influenced by the biological predispositions that are most necessary for safe movement [88]. Similarly, motor-cognitive DT aims to enhance balance and reduce falls in both domains. Mirelman et al. postulated that combining a treadmill exercise with a cognitive exercise would mitigate fall risk significantly during the six months after training [56]. Integrated DT allows tasks to become habitual and minimizes competition to improve balance.

The reduced resource theory explains that the repetitions in DT training increase cognitive capacity, which in turn improves balance [62]. The task coordination theory describes the coordination and management of DT performance to optimize stability during the performance of concurrent tasks [89]. The capacity-sharing theory states that effective DT integration promotes DT performance by sharing brain resources between tasks [90,91]. In other words, sharing capacity improves through task integration, as brain resources are shared between two tasks, resulting in better DT performance [90]. According to the limited resource theory, the tasks in DT training compete for limited neural resources [91]. After performing the DT exercise, DT performance improves because each task becomes automatic and competition reduces [91]. Thus, complex motor-cognitive training makes tasks automatic and less competitive, thereby improving balance control in older adults [61].

## 5. Implications

The findings of this review provides a better understanding of the underlying theories for improvement following DT training. Our Table 2 reports a summary of the type of exercise, dosage of intervention, outcomes on balance and falls reduction and the proposed theory underlying the improvement. These findings benefit researchers by helping them to plan future intervention studies that could bridge the literature gap such as including cognitive–cognitive DT exercises and for the clinicians, these findings will assist in making a clear choice on the type and dosage of DT intervention for achieving specific health benefit among the older adults.

This review has several limitations. (1) Most of the studies did not mention the repetition and allocated time for DT separately (i.e., the dosage for the primary and secondary tasks). The load and complexity of the exercise, as well as the clinician’s skills, are key to achieving a successful outcome from intervention; however, they were addressed inadequately in the included studies. Future RCTs should address these inadequacies appropriately. (2) Our included studies were restricted to English and Chinese language publications; therefore, it is possible that potentially relevant studies [92,93] were not considered. (3) The findings of this study must be interpreted with caution due to the heterogeneity in participant characteristics, study methods, type of DT intervention, outcome measures used, dosage of intervention, study setting and mode of intervention delivery. (4) More than half (60%) of the studies were classified as having poor methodological quality, and as quality impacts the study results, this must be considered when interpreting the findings. (5) Not all RCTs necessarily support the theories. Among the potential studies for inclusion, two studies [42,43] that did not record an improvement in balance or fall reduction did not report a theory to support and therefore they were excluded in the full-text screening stage. In addition, one study [60], though did not record an improvement mentioned a theory for supporting the benefits of the intervention. (6) We restricted studies among healthy older adults without pathological conditions and therefore generalizing these findings to all older adults is limited. Future reviews are warranted to study special groups of older adults with pathological conditions and (7) we did not attempt to explore the reasons for improvement among the studies that did not report the theory (*n* = 2). This strategy is in line with our review protocol. This review also has the following merits: (1) extensive searches for relevant studies were performed across 11 databases using a systematic methodology and (2) since this review exclusively included RCTs, our conclusions are based on high-quality evidence.

## 6. Conclusions

This review identified 20 possible theories to explain the improvement in balance and reduction in falls following DT training among older adults. The dominant theory invoked to explain the effectiveness of DT training was the attention theory, which is commonly proposed in motor-cognitive DT studies. In motor-cognitive DT training, attention is shifted based on the complexity and priority of a task from the motor to the cognitive task to improve balance and reduce falls incidence among older adults.

## Figures and Tables

**Figure 1 ijerph-19-16890-f001:**
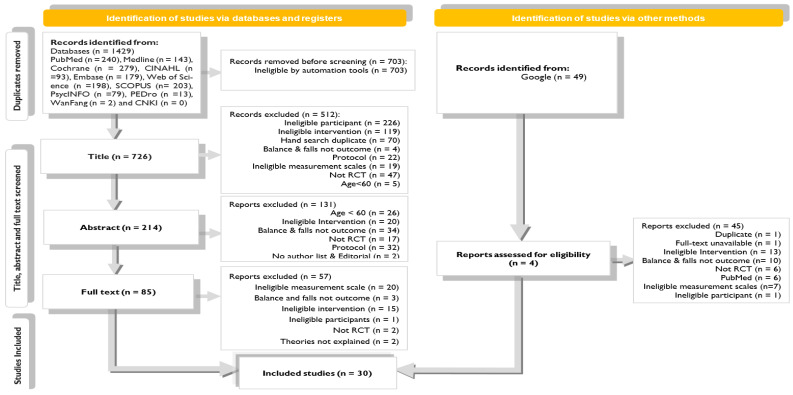
Flowchart of study inclusion.

**Figure 2 ijerph-19-16890-f002:**
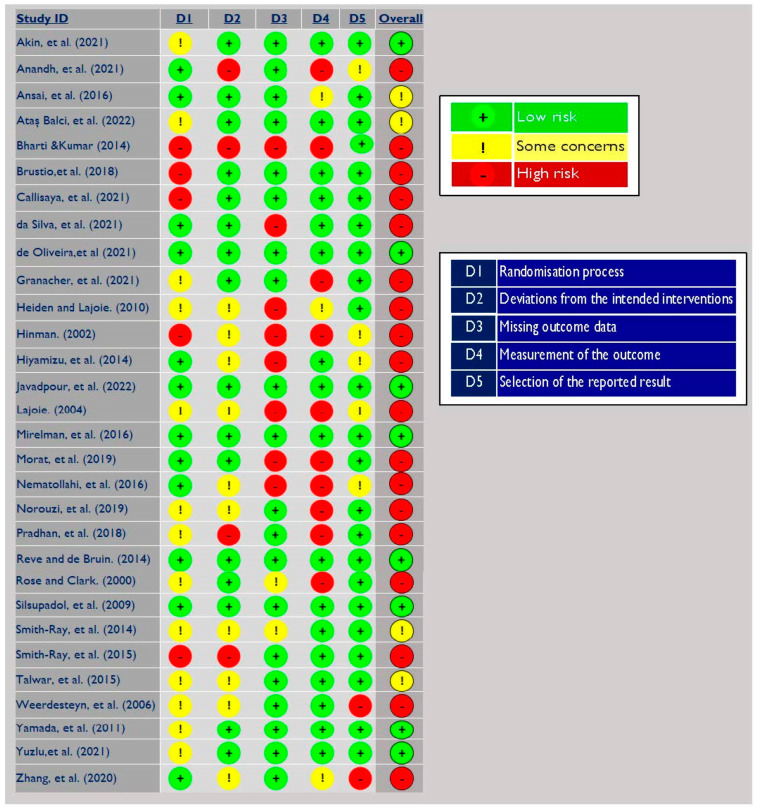
Risk of bias analysis of the included studies [21,22,23,28,29,30,44,45,46,47,48,49,50,51,52,53,54,55,56,57,58,59,60,61,62,63,64,65,66,67].

**Figure 3 ijerph-19-16890-f003:**
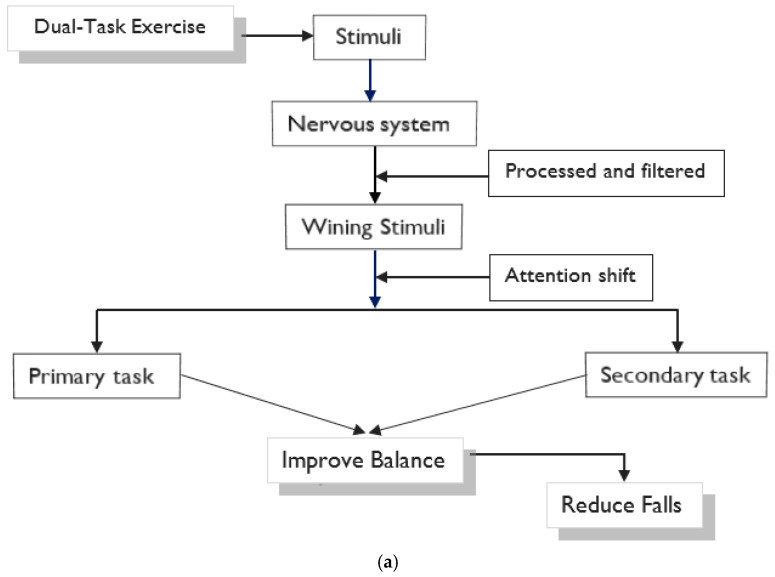
(**a**) Concept of neurophysiological phenomenon of attention theory for DT intervention; (**b**) Concept of executive function approach in application of DT Intervention; (**c**) Concept of Multicomponent Training Approach of DT Intervention; (**d**) Concept of Working Memory Model in Application of DT Intervention.

**Table 1 ijerph-19-16890-t001:** Dual-task training classification, definitions, and examples of exercises.

DT Training/ Classification	Definition	Exercise Reported in Trials
Motor-motor DT training	Exercise is assigned to retain postural control of the body by employing both motor and motor tasks simultaneously.	Walk and hold two half-filled glasses in both hands; Daily brushing with balance exercise
Motor-cognitive DT training	Exercise is intended to develop the abilities of posture control and executive function through the synchronous performance of one motor and one cognitive task.	Balance exercise with a verbal fluency task; Squire-stepping exercises in fixed-priority and variable-priority instruction
Cognitive–cognitive DT training	Exercise training with both cognitive tasks performed at the same time is intended to improve the executive function of the brain.	In a computer game, players see an animated truck and a road sign in the background before they fade away. Subsequently, two vehicles reappear to identify the correct vehicle. Meanwhile, a circle of cars appears around the edge, with one road sign. Participants need to find the location where the road sign first appeared on the edge.

**Table 2 ijerph-19-16890-t002:** Dual-task training and the proposed theory from all included trials.

Authors/ Year	Population/Gender/ Study Setting/ Country	Mean Age/Age Range/Sample Size	Intervention (Dual-Task)				Assessment Time	Control	Effectiveness of Intervention		Authors Conclusions	Proposed Theory
			Primary Task	Secondary Task	Dosage	Total Time			Balance	Falls		
			(Rep/Minutes/Task/Load/Complexity/Administer)		(Duration/Week/Total)	(DT)						
Akin et al. (2021) [28]	-/ 13 male & 19 female/ Laboratory/ Turkey	67.72/ -/ 50 (Intervention G1: 25/Intervention G2: 25)	G1: Motor training G2: Motor training (20 min/Administer: Physiotherapist)	G1: Cognitive Training G2: Motor Training (20 min/Administer: Physiotherapist)	40 min/ -/ 8 weeks	20 min	Pre and post		Intervention: G1: ↑↑↑ BBS G2: ↑↑↑ BBS		Both programs improved balance, fear of falling, walking, and muscle strength.	Attention theory
Anandh et al. (2021) [44]	Community dwellers /-/ Medical institute/ India	-/ 65–75/ 96(-/-)	G1: Motor training on regular surface G2 Motor training on uneven surfaces (15 min/Load: ↑/Administer: Physiotherapist)	G1&G2: Cognitive training (15 min/Load: ↑/Administer: Physiotherapist)	60 min/ -/ 1 year	G1: 5 min G2: 10 min			Intervention: G1&G2: ↑↑↑ TBAT		Preventing secondary impairments Introduce progressive, safe, dual-task activities on even regular surfaces and multi-task conditions for further progression.	Attention theory
Ansai et al. (2016) [67]	Community dwellers/ 47 female and 22 male/ University/ Brazil	82.4/ -/ 69 (Intervention G1: 23/Intervention G2: 23/Control: 23)	G1: Motor training G2: Motor training (Repetation: 10–12/3 min/Load: ↑&↓/Physical educator)	G1: Motor/cognitive Training (53 min/Load: ↑/Complexity: ↑/Administer: Physical educator)	60 min/ 3 times/ 16 weeks		Pre, post, and retention (6 weeks)	No intervention	Intervention and Control: ↑↑ OLS ↓↓ TT	Intervention G1: ↓↓ frequency	With higher adherence to protocol, multicomponent training is more effective and presents fewer adverse events.	Multicomponent training approach
Ataş Balci et al. (2022) [63]	Community dweller/ 39 female and 6 male/ University/Turkey	73.0/ / 45(Intervention G1: 15/Intervention G1: 15/Control G3: 15)	G1: Motor training G2: Motor training G3: Motor training (Minutes: ↑/Load: ↑/Complexity: ↑/Administer: Physiotherapist)	G2: Cognitive training G3: Cognitive training (Successive) (Minutes: 30/Complexity: ↑/Administer: Physiotherapist)	G1&G2: 30 min G3: 60 min/ 3 times/ 4 weeks	30 min	Pre and post-test		G1: ↑↑↑ BBS G2: ↑↑↑ BBS G3: ↑↑↑ BBS		Successive physical-cognitive training is more successful at improving balance and reducing fall fear in the elderly.	Capacity-sharing theory
Bharti &Kumar (2014) [53]	Residencial care dweller/ -/ India	73.63/ -/ 30 (Intervention: 15/Control: 15)	G1: Motor training G2: Motor training (Complexity: ↑)	G1: Cognitive training (Variable priority) G2: Cognitive training (Fixed priority)	45 min/ 3 times/ 4 weeks		Pre and post-test		G1&G2: ↑↑ TPOMA		Balance of older adults improves after dual-task training with fixed and variable priorities.	Task coordination theory
Brustio et al. (2018) [64]	Private senior social center/ 18 male & 42 female /-/ Italy	73.5/ 70–80/ 60 (Intervention G1: 19, Intervention G2: 19^/^Control: 22)	G1: Motor training (Repetition: 12/60 min/Complexity: ↑or↓) G2: Motor training (60 min/Complexity: ↑)	G1: Motor training (31 min/Complexity: ↑)	60 min/ 2 times/ 16 weeks	31 min	Pre and post-test	Usual care	Intervention: G1: ↑↑↑ FSST G2: NS FSST Control: NS FSST		Motor DT training incorporates motor extra activities and has the potential to improve mobility.	Attention theory
Callisaya et al. (2021) [54]	Community dwellers &clinics/ 39 male & 54 female/ Home/ Australia	72.8/ -/ 93 (Intervention: 17 and Control: 22)	Motor training (Repetition: 3/40–120 min/Load: ↑/Complexity: ↑)	Cognitive training (Repetition: 3/10–30 min/Load: ↑/Complexity: ↑)	40–120 min/ weeks/ 24 weeks		Pre and post test	No intervention	Intervention and Control: NS ↑↑ ST NS ↑↑ FICSIT-4		Trend towards enhanced gait speed.	Executive function theory Processing speed theory
da Silva et al. (2021) [29]	Community dweller/ 14 female and 2 male/ University clinic/ Brazil	71.5/ -/ 16 (Intervention: 10/Control: 6)	G1: Motor training G2: Motor training (Repetation: 12/40 min/Complexity: ↑or↓)	G1: Cognitive (Repetation: 12/Complexity: ↑↓)	-/ 3 times/ 6 weeks		Pre, second and post-test		Intervention: G1&G2: ↑↑↑ F8W		Hemodynamic stability, comprehension and adherence to interventions, increased mobility and frailty, static postural control, and dynamic balance.	Multicomponent training approach
de Oliveira et al. (2021) [55]	Community dweller/ -/ -/ Brazil	68.3/ -/ 50 (Intervention: 25/Control: 25)	Motor training (Repetation: 7–12/Load: ↑)	Cognitive Training (Reputation: 1–5/Load: ↑)	60 min/ 3 times/ 24 weeks		Pre, second, and post	Motor training (Repetation: 7–12/Load: ↑)	Intervention: ↑↑↑ BBS Control: ↑↑ BBS		SRT outcomes were better in the UST group, while C + UST resulted in greater gains in the TUG test.	Executive function theory Multicomponent training approach
Granacher et al. (2021) [45]	Community dweller/ 27 female and 24 male/Laboratory/Germany	65.65/ 60–72/ 51(Intervention: 27/Control: 24)	Motor training (3 min)	Motor training (Load: ↑/Complexity: ↑)	3 min/ 2 times/ daily/ 8 weeks	336 min for 112 sessions	Pre and post-test	No intervention	Intervention: ↑↑↑ TUG ↑↑↑ FRT ↓↓RT Control: ↓↓ TUG ↓↓ FRT ↓↓ RT		Lifestyle balance training program during tooth brushing is insufficient to improve balance and muscle strength.	Attention theory
Heiden, and Lajoie. (2010) [48]	Community dweller/ 11 female and 5 male/ -/ Canada	77/ -/ 16(Intervention: 9/Control: 7)	Motor training (30 min/Instructor)	Cognitive training	30 min/2 times/ 8 weeks		Pre-post and retention test (2 weeks)	Motor training (60 m/ 2 times/ 8 weeks)	Intervention: ↑↑↑ CB&M		Games-based balance biofeedback training significantly improves functional balance by reducing the attentional demands of postural control.	Attention theory
Hinman et al. (2002) [50]	Community dweller/Home and laboratory/55 female and 33 male/ USA	72/ 63–87/ 88 (Intervention: 28/Intervention: 30/Control: 30^)^	G1: Motor training (Complexity: ↑) G2: Motor training (10 min/ Load: ↓/Complexity: ↑)	G2: Cognitive training	20 min/ 3 times/ 4 weeks		Pre and post	No intervention	Intervention: G2: ↑↑ BBS G1: ↑↑ BBS Control: ↑↑ BBS		Greater degree of impaired participants who received intense training beyond 4 weeks mostly benefited.	Multicomponent training approach Low-tech approach. Whipple’s concentration approach.
Hiyamizu et al. (2012) [49]	Community dweller/ 26 female and 10 male/ Japan	71.6/ -/ 43(Intervention: 21/Control: 22)	Motor training (Administer: Therapist)	Cognitive training	60 min/ 2 times/ 12 weeks		Pre and post	Motor training	Intervention and Control: NS ↑↑ TUG		Dual task balance training improves standing postural control in the elderly.	Working Memory model
Javadpour et al. (2022) [21]	Community dweller/ 49 female and 20 male /-/ Iran	68.6/ 65–79/ 69(Intervention G1: 23 and G2: 23, and control: 23)	G1: Motor training G2: Motor training (Administer: Physiotherapist)	G2: Cognitive training	40–60 min/ 3 times/ 6 weeks		Pre and post	No intervention	InterventionG1&G2: ↑↑↑FAB Control: NS FAB		Balance training enhance gait smoothness and balance in healthy older persons.	Working memory model Task oriented approach Goal oriented approach
Lajoie. (2004) [51]	Community dweller/ 20 female and 4 male s/-/ Canada	70.85/ -/ 24 (Intervention: 12/Control: 12)	Motor training (Repetatopn: 15/1 min/Complexity: ↑)	Cognitive training	60 min/ 2 times/ 8 weeks		Pre, post and retention (2 weeks)	No intervention	Intervention: ↑↑↑ BBS Control: NS BBS	↓↓%	Automaticity of maintaining a static posture increases significantly after postural training with feedback fading protocol.	Attention theory
Mirelman et al. (2016) [56]	Community dweller/ 100 female &182 male/ Clinic/Belgium, Israel, Italy, Netherlands, and UK	73.75/ 60–90/109 (Intervention: 52/Control: 57)	Motor training (Load: ↑/Complexity: ↑)	Cognitive training (Load: ↑/Complexity: ↑)	45 min/ 3 times/ 6 weeks		Pre, post and retention (24 weeks)	Motor training (45 min/ 3 times/ 6 weeks)		↓↓↓ Incidence rate	Treadmill with virtual reality training resulted in lower fall rates.	Integrated motor and cognitive theory
Morat et al. (2019) [52]	Community dweller/ 17 male and 28 female/ -/ Germany	69.4/ -/ 45(Intervention G1: 15/Intervention G1: 15/Control: 15)	G1: Motor training G2: Motor training (Load: ↑/Administer: Study assistant)	G1: Cognitive Training (Unstable surface) G2: Cognitive Training (10–12 min//Study assistant)	40 min/ 3/ 8 weeks		Pre and post-test	Maintain a level of activity	Intervention: G1: ↑↑↑ Y-BT ↑↑↑TUG G2: ↑↑↑ Y-BT ↑↑ TUG Control: ↑↑ Y-BT ↓↓ TUG		Under stable and unstable situations, volitional stepping exergames are an effective training method with excellent adherence rates for improving functional balance and calf strength.	Divided attention theory Selective attention theory Cognitive flexibility theory Working memory model
Nematollahi et al. (2016) [22]	Community dweller/ 12 male and 32 female/ -/ Iran	66.4/ 60–70/ 57(Intervention G1: 19/Intervention G2: 19/Control: 19)	G1: Motor training G2: Motor training G3: Motor training (55 min/Administer: Physiotherapist)	G1: Cognitive training G2: Cognitive training (40 min/Administer: Physiotherapist)	60 min/ 3 times/ 4 weeks		Pre and post-test		Intervention and Control: ↑↑↑ FAB		Traditional, multisensory, and dual-task balance training is beneficial for improving balance, with no clear advantage over the others.	Attention theory
Norouzi et al. (2019) [23]	-/ 60 male/ 65–75/ Iran	68.31/ -/60 (Intervention: 20/Intervention: 20/Control: 20)	G1: Motor training G2: Motor training (Repetation: 8/60–80 min/Load: ↑/Physiotherapist)	G1: Motor training (60–80 min/Physiotherapist) G2: Cognitive training	60–80 min/ 3 times/ 4 weeks		Pre, post, and retention (12 weeks)	Informal meeting+ Maintain physical activities+ Refrain from sports activities	Intervention G1: ↑↑ BBS G2: ↑↑↑ BBS Control: ↓↑ BBS		Working memory and balance performance improved more with mCdtt than with mMdtt.	Divided attention theory Attention theory Executive function theory Working memory model
Pradhan et al. (2018) [46]	Community dweller/ 18 male and 22 female/ -/ India	69.75/ 65–75/ 40(Intervention: 20/Intervention: 20)	Motor training (Load: ↑/Complexity: ↑)	G1: Cognitive training (40 min/Complexity: ↑)	45 min/ 3 times/ 4 weeks		Pre and post test	Walk	Intervention: ↑↑↑ BBS and ↑↑↑ TUG Control: ↑↑↑ BBS & ↑↑↑ TUG		Multiple-task exercises combined with cognitive tasks improve gait balance by keeping people awake and attentive while walking.	Attention theory
Reve and de Bruin (2014) [57]	Homes-for-the-aged and community dweller, vicinity homes/ -/Switzerland, Germany	81.5/ -/ 182(Intervention: 94/Control: 88)	G1: Motor training G2: Motor training (40 min/Load: ↑or↓)	G2: Cognitive training	40 min/ 2 times/ 12 weeks	10 min/ 3 times/ weeks	Pre, post and retention test (48 weeks)			↓↓ %	DT costs of walking, gait initiation, and divided attention are reduced when strength-balance and cognitive training are combined. Strength-and-balance training enhances executive functioning lowering the risk of falling.	Executive function theory Divided attention theory
Rose, and Clark (2000) [47]	Community dweller/ 28 female and 13 male/Laboratory /USA	78.7/ 72–85/ 45 (Intervention: 24/Control: 21)	Motor training (Load: ↑/Complexity: ↑)	Motor training (Load: ↑/Complexity: ↑)	45 min/ 2 times/ 8 weeks		Pre and post test	No intervention+ Not alter physical activities	Intervention: ↑↑ BBS ↑↑ TUG. Control: ↑↑ BBS ↑↑ TUG		Theory-driven rehabilitation programs significantly improve the control of bodily orientation in both static and dynamic action environments.	Theory of perception and control body orientation.
Silsupadol et al. (2009) [58]	Community dwellers/ 18 female and 5 male/ Laboratory/ USA	74.8/ 65–85/ 23 (Intervention G1: 8/Intervention G2: 8/Intervention G3: 7)	G1: Motor training G2: Motor training G3: Motor training (Repetation: 4/12 min/Load: ↑/Complexity: ↑/Physiotherapist&2 Research assistance)	G2 + G3: Cognitive training	45 min/ 3 times/ 4 weeks		Pre and post		InterventionG1,G2&G3: ↑↑ BBS		Improves balance in people with balance impairments. Single-task is unable to transfer to dual-task balance control.	Attention theory Divided attention theory Task integration model. Task-automation model
Smith-Ray et al. (2014) [66]	Community dwellers/ 42 female and 4 male/ Community center/ USA	72.47/ 66–79/ 45 (Interventon: 23 and Control: 22)	Cognitive exercise (3 Computer games)		60 min/ 2 times/ 10 weeks		Pre and post-test	No intervention	Intervention: ↑↑↑ BBS Control: ↑↑ BBS		Study presents preliminary evidence that cognitive training improves balance and mobility in older adults who have a history of falls.	Executive function theory Attention theory
Smith-Ray et al. (2015) [65]	Independent living facilities/ 39 female &12 male /-/ USA	81.9/ 75–88.3/ 53(Intervention: 27 and Control: 24)	Cognitive exercise (3 Computer games)		60 min/ 3 times/ 10 weeks	IMPACT: 2400 min ACTIVE: 750 min	Pre and post-test	2 fall prevention brochures	Intervention: ↑↑↑ TUG Control: ↑↑↑ TUG		Cognitive training is a potential method to fall prevention.	Executive function
Talwar et al. (2015) [59]	Old-age home/ -/-/ India	76.4 65–89/60/(20 and Intervention: 20 and Intervention: 20) /(Intervention G1: 20 Intervention G2: 20 and Intervention G3: 20)	G1: Motor training G2 + G3: Motor training (Complexity: ↑/Administer: Physiotherapist)	G2: Motor training G3: Motor training	60 min/ 3 times/ 4 weeks		Pre and post-test		Intervention G1, G2&G3: ↑↑↑ BBS		Agility training improves balance in people with balance problems.	Attention theory
Weerdesteyn et al. (2006) [30]	Community-dweller/ 82 female and 25 male/ Laboratory/ Netherlands	73.93/ -/ (Intervention G1: 49/Intervention G2: 30/Control: 28)	Motor training (1st and 2nd session)	Cognitive training (1st session) Motor training (2nd session)	90 min/ 2 times/ 5 weeks		Pre and post-test	No intervention		Intervention: ↓↓ IRR	Intervention was effective in reducing the incidence of falls.	Multicomponent training approach
Yamada et al. (2011) [60]	Community dweller/ 15 male and 38 female /-/ Japan	80.8/ 67–97/ 53(Intervention G1: 26/Intervention G2: 27)	G1: Motor training (Repetaion: 100/5 s/Load: ↑/Administer: Physiotherapist) G2: Motor training (Repetation: 100/10 s/Administer: Physiotherapist)	G1: Cognitive training	50 min/ 1 time/ 24 weeks		Pre and post-test		Intervention G1&G2: ↓↓ TUG		Balance did not improve rather improving ambulatory function.	Attention theory
Yuzlu et al. (2021) [61]	Community dweller/ 11 male & 47 female/ Private institution/Turkey	84.1/ -/ 58 (Intervention G1: 29 and Intervention G2: 29)	G1: Motor training G2: Motor training (40 min/Load: ↑/Complexity: ↑/Administer: Physiotherapist)	G1&G2: Cognitive training (G1: Combine &G2: Subsequently)(40 min/Complexity↑/Physiotherapist)	60 min/2 times/ 8 weeks		Pre and post-test		InterventionG1 and G2: ↑↑↑ BBS, ↑↑↑ TUG		Integrated and consecutive DT balance training on balance and gait performance were not statistically significantly different.	Capacity-sharing theory Limited resource theory Task integration model. Task-automation model
Zhang et al. (2020) [62]	Geriatric nursing home/ 12 male & 18 female/ -/China	83.27 /-/ (Intervention: 15/Control: 15)	Motor training (Complexity: ↑)	Cognitive training (Complexity: ↑)	30 min/ 5 times/ 6 weeks		Pre &post test	Motor training	Intervention: ↑↑↑ TUG Control: ↑↑ TUG		Significantly improve the stride length and cadence. The effect lasts longer and requires less energy.	Theory of reduced resources allocation Attention theory

↑ = Increased, ↓ = Decreased, ↑↑ = Improved, ↓↓ = Reduced, ↑↑↑ = Significantly Improved, ↑↓ = Remain Same, - = Not Reported, NS = Non-significant. G1 = Group 1, G2 = Group 2, IMPACT = Cognitive Training Improvement in Memory with Plasticity-based Adaptive Cognitive Training, ACTIVE = Advanced for Independent and Vital Elderly, BBS = Berg Balance Scale, TUG = Timed Up and Go, CB&M = Community Balance and Mobility Scale, OLS = One leg stance, TT = Tandem test, RT = Romberg test, ST = Step test, FAB = Fullerton Advanced Balance, FSST = Four square step test, F8W = Figure of 8 Walk test, FICSIT-4 = Frailty and Injuries Cooperative Studies of Intervention Techniques test, Y-BT = Y-balance test, TBAT = Tinetti Balance Assessment Tool, Balance Exercis, TPOMA = Tinnetti Performances Oriented Mobility Score, FRT = Functional reach test.

## Data Availability

Not applicable.

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
