# Peer review of "A Systematic Review Exploring the Theories Underlying the Improvement of Balance and Reduction in Falls Following Dual-Task Training among Older Adults"

_ijerph, 2022, doi:10.3390/ijerph192416890_

Round 1

Author Response

Dear Reviewer,

Thank you for contributing to our articles to improve the quality of the manuscript.

Please find the manuscript and reviewer comments in separate files.

Changes made in response to the comments are noted in yellow in the manuscript, and italics and “double quotes” are used in the reviewer's comments file.

Thank you!

Regards,

Mohammad Jobair Khan

Reviewer 2 Report

Thank you for the opportunity to review this submitted manuscript.

This systematic review aims to explore the mechanisms underlying the effectiveness of dual-task training in improving balance and reducing falls in elderly. It adheres to the recommendations of the PRISMA statement and thoroughly evaluates the risk of bias (RoB) in the included studies using the Cochrane Collaboration RoB 2 tool.

20 possible mechanisms could be identified to explain the improvement in balance and reduction in cases following DT training among older adults.

Multiple previous systematic reviews relating to dual task training can be found by a quick search on pubmed, most of them with a similar setting, however, commonly with a focus on specific diseases (Stroke, Parkinson's, multiple sclerosis,...). To be included in the present review, the studies had to have investigated older adults (aged > 60 years) without any pathological conditions.

The authors have made an effort in their literature research and searched through all the major databases.

With regard to the graphical processing of the selection process, minor errors are noticeable. The following calculation should be corrected: "Title (n = 726)" followed by "Records excluded (n = 488)", with actually 512 exclusions being named in detail, which are further required for the calculated "Abstract (n = 214)". In the Google column, the "Reports not retrieved (n = 44)" should consequently appear next to the first field.

A prior systematic review by Agmon in 2014 (which the authors also shortly quote) had also set itself the task of 1) identifying clinical or community-based interventions that improved dual-task postural control among older adults; and 2) to identify the key elements of those interventions. Twenty-two studies were found by the end of 2013 (several identical with the present study), which can now be expanded with the present research.

As with the first study, it must also be noted that the validity of the evaluation is made more difficult by the heterogeneity in participant characteristics, study designs, and outcome measures. This should be stated more clearly in the limitations.

In summary, even if the subject of the review is not entirely new. a high-quality study is presented here. The introduction is thorough, the methodology is solid, and the discussion of the results follows a largely comprehensible common thread. I recommend acceptance of the manuscript after minor revision.

Author Response

(The authors gave the same response as above.)

Round 2

Author Response

Dear Reviewer,

Thank you for your kind review. Please find the response in the enclosed file.

Regards,

Jobair
